# Effects of Quercetin-Loaded Nanoparticles on MCF-7 Human Breast Cancer Cells

**DOI:** 10.3390/medicina55040114

**Published:** 2019-04-22

**Authors:** Firoozeh Niazvand, Mahmoud Orazizadeh, Layasadat Khorsandi, Mohammadreza Abbaspour, Esrafil Mansouri, Ali Khodadadi

**Affiliations:** 1Cellular and Molecular Research Center, Ahvaz Jundishapur University of Medical Sciences, Ahvaz, Iran; fniazvand@gmail.com (F.N.); orazizadehm@gmail.com (M.O.); mansouri-e@ajums.ac.ir (E.M.); 2Department of Anatomical Sciences, Faculty of Medicine, Ahvaz Jundishapur University of Medical Sciences, Ahvaz, Iran; 3Targeted Drug Delivery Research Center, School of Pharmacy, Mashhad University of Medical Sciences, Mashhad, Iran; abbaspourmr@mums.ac.ir; 4Cancer Research Center, Ahvaz Jundishapur University of Medical Sciences, Ahvaz, Iran; akhodadadi@ajums.ac.ir; 5Department of Immunology, Faculty of Medicine, Ahvaz Jundishapur University of Medical Sciences, Ahvaz, Iran

**Keywords:** apoptosis, toxicity, quercetin, solid lipid nanoparticles, breast cancer, nanomedicine

## Abstract

*Background and objectives*: Previous studies have shown anti-tumor activity of quercetin (QT). However, the low bioavailability of QT has restricted its use. This study aimed to assess the toxic effect of QT encapsulated in solid lipid nanoparticles (QT-SLNs) on the growth of MCF-7 human breast cancer cells. *Materials and Methods*: MCF-7 and MCF-10A (non-tumorigenic cell line) cell lines treated with 25 µmol/mL of QT or QT-SLNs for 48 h. Cell viability, colony formation, oxidative stress, and apoptosis were evaluated to determine the toxic effects of the QT-SLNs. *Results*: The QT-SLNs with appropriate characteristics (particle size of 85.5 nm, a zeta potential of −22.5 and encapsulation efficiency of 97.6%) were prepared. The QT-SLNs showed sustained QT release until 48 h. Cytotoxicity assessments indicated that QT-SLNs inhibited MCF-7 cells growth with a low IC_50_ (50% inhibitory concentration) value, compared to the free QT. QT-SLNs induced a significant decrease in the viability and proliferation of MCF-7 cells, compared to the free QT. QT-SLN significantly increased reactive oxygen species (ROS) level and MDA contents and significantly decreased antioxidant enzyme activity in the MCF-7 cells. Following QT-SLNs treatment, the expression of the *Bcl-2* protein significantly decreased, whereas Bx expression showed a significant increase in comparison with free QT-treated cells. Furthermore, The QT-SLNs significantly increased apoptotic and necrotic indexes in MCF-7 cells. Viability, proliferation, oxidative stress and apoptosis of MCF-10A cells were not affected by QT or QT-SLNs. *Conclusions*: According to the results of this study, SLN significantly enhanced the toxic effect of QT against human breast cancer cells.

## 1. Introduction

Breast cancer is one of the most diagnosed cancers worldwide. There are several types of therapy for breast cancer, such as radiotherapy and chemotherapy. However, these therapies have several side effects on healthy cells [1,2,3]. Recently, many studies have focused on finding new drugs to treat breast cancer [4,5,6,7]. Some phyto-bioactive compounds act as pro-oxidants that cause reactive oxygen species (ROS) in cancer cells [8,9]. Oxidative stress can alter signaling pathways, damages the DNA, and affect progress various cancers, such as colon, prostate, lung, ovary, and breast [10]. Superoxide dismutase (SOD) and Catalase (CAT) are major enzymatic antioxidants involved neutralize ROS in cancer cells [11].

Quercetin (QT), a powerful flavonoid, is found in onion, red grapes, lettuce, tomato, olive oil, tea, coffee, bracken fern, and citrus fruits [12]. QT has toxic effects on many types of cancer cells [13,14,15]. However, poor solubility and low bioavailability of QT have limited its therapeutic applications [15,16,17]. Several drug delivery systems, such as polymeric nanoparticles, solid lipid nanoparticles (SLNs), liposomes, and micelles, have examined to increase the bioavailability of anticancer agents [18,19,20,21]. 

Recently, the conjugates of bioflavonoids and nanoparticles have studied for targeted drug delivery systems [22,23]. However, the safety of these nanoparticles is highly controversial [24,25,26].

In recent years, SLNs have extensively used as a carrier for various anticancer drugs and phyto-bioactive compounds [21,27]. For example, Tamoxifen-loaded SLNs suppress breast tumors in rats [25], and pomegranate extract-loaded SLNs effectively prevent the proliferation of various cancer cells such as PC-3, MCF-7 and HepG2 [28]. The SLNs have a high drug-loading capacity, increase the blood circulation time, modulate release kinetic, increase the therapeutic efficacy of anticancer drugs, and protects the encapsulated compound from chemical degradation [29,30,31].

Until now, the anticancer effects of various natural phyto-bioactive compounds loaded with SLNs have been investigated in several studies [32,33,34], but less attention has been paid to their cell death mechanism. The major problem of cancer therapy is the ability of cancer cells to evade apoptosis, resulting in resistance to treatment. Therefore, developing new therapeutic agents to overcome treatment resistance is of paramount importance. This study aimed to investigate the cytotoxic and apoptotic effects of QT-SLNs on MCF-7 cell line.

## 2. Materials and Methods

### 2.1. Preparation of QT-SLNs

The SLNs of QT were prepared using Compritol (as lipid, Gattefossé, Saint-Priest, France) and Tween 80 (as a surfactant) through a microemulsification technique [35,36]. Briefly, Compritol was heated to 70–75 °C, and 50 mg of QT (Sigma-Aldrich, St. Louis, MO, USA) was added to the molten lipid. Six mL of water and Tween 80 were mixed separately and heated to 70–75 °C. Then, the two solutions were mixed and stirred to produce a clear homogenous microemulsion. The homogenized microemulsion was added to the 100 mL of cold water and stirred (40 min) to get a fine dispersion of the SLNs. The QT-SLNs suspension was stored at 4 °C. Different formulations of QT-SLNs based on lipid-drug ratios were prepared (Table 1). 

### 2.2. Particle Size and Zeta Potential of QT-SLNs

The average particle size distribution, polydispersity index (PDI), and zeta potential of the QT-loaded SLNs were assessed using dynamic light scattering (DLS) method by a Zetasizer-Nano-ZSP (Malvern, UK). The particle morphology was also evaluated by a transmission electron microscope (TEM) (JEOL Ltd., Tokyo, Japan).

### 2.3. Drug Entrapment Efficiency

The prepared QT-loaded SLNs were centrifuged at 14,000 rpm for 7 min to separate non-entrapped QT. The supernatant was analyzed spectrophotometrically at 256 nm (UV 1700, Shimadzu, Kyoto, Japan) for detection of the QT amounts. Encapsulation efficiency (EE) was calculated using the following formula [36,37]. 

(1)$$\%\;\mathrm{EE}\;=\frac{\mathrm{Amount}\;\mathrm{of}\;\mathrm{drug}\;\mathrm{added}-\mathrm{Amount}\;\mathrm{of}\;\mathrm{drug}\;\mathrm{in}\;\mathrm{the}\;\mathrm{supernatant}}{\mathrm{Amount}\;\mathrm{of}\;\mathrm{drug}\;\mathrm{added}}\times \;100$$

To determine the drug loading (DL), 50 mg QT powder was extracted by using absolute methanol. The QT extracted was then diluted up to 10 mL, and QT contents of the solution were analyzed spectrophotometrically at 256 nm. The DL percentage was calculated using the following formula [37].

(2)$$\%\;\mathrm{DL}\;=\;\frac{\mathrm{Weight}\;\mathrm{of}\;\mathrm{initial}\;\mathrm{drug}-\;\mathrm{Weight}\;\mathrm{of}\;\mathrm{free}\;\mathrm{drug}}{\mathrm{Weight}\;\mathrm{of}\;\mathrm{lipid}}\;\times \;100$$

### 2.4. In Vitro Drug Release

QT release from SLNs was measured using the dialysis bag method, as previously described [36,37]. An accurately weighed amount of QT-loaded SLN dispersions containing the drug equivalent to 3 mg was poured into a dialysis bag (Viskase, Lombard, IL, USA). The dialysis bag retains nanoparticles and allows the diffusion of the free drug into dissolution media (PBS at pH: 7.4). At predetermined time intervals, samples were withdrawn and analyzed spectrophotometrically (UV 1700, Shimadzu, Japan) at 256 nm. The following formula was used to calculate the percentage of drug release [36].

(3)$$\%\mathrm{Drug}\;\mathrm{released}\;=\frac{\mathrm{Released}\;\mathrm{QT}}{\mathrm{Total}\;\mathrm{QT}}\;\times \;100$$

### 2.5. Experimental Design

The human MCF-7 and MCF-10A cell lines were purchased from the National Center for Genetic and Biological Reserves in Iran and cultured in DMEM/F12 medium supplemented with 10% FBS, streptomycin (100 U/mL) and penicillin (100 mg/mL). The cells were maintained in a humidified atmosphere of 5% CO_2_ at 37 °C. The cells were categorized into four groups as follows:Control: received only mediaBlank SLN: exposed to 25 µmol/mL of SLN without QTQT: treated by 25 µmol/mL of QTQT-SLN: treated with 25 µmol/mL of QT-SLNs


The Ethics committee of the Ahvaz Jundishapur University of Medical Sciences approved this study (approved number: IR.AJUMS.REC.1394.9418). The dose and exposure time of QT and QT-SLNs were selected based on 50% inhibitory concentration (IC50) value (Table 2). The MCF-7 and MCF-10A cells were treated with IC50 concentrations of QT-SLNs. QT was dissolved in 1% Dimethyl sulfoxide (DMSO) and diluted in culture medium. To evaluate the safety of the DMSO, the MCF-7 and MCF-10A cells were treated with QT (40 µM) or 1% DMSO for 48 h, and the MTT assay was performed. DMSO had no significant effect on the viability of the MCF-7 and MCF-10A cells (Appendix A).

### 2.6. Cell Viability 

MTT assay was used to compare the effect of QT-SLNs with QT on cell viability. Briefly, MCF-7 and MCF-10A cells (1 × 10^4^ cells/well) were cultured in 96-well plates. After treatment, the MTT solution at a concentration of 0.5 mg/mL was added to each well and maintained at 37 °C for 4 h. After removing the supernatants, 100 µL of DMSO was added to each well. Using a microplate reader (BioRad, Hercules, CA, USA), absorbance at 570 nm was measured. To determine the toxic effect of QT-SLNs on the MCF-7 cells, IC_50_ values were measured by MTT assay, as previously described [38]. The IC_50_ values were calculated using SigmaPlot software. 

### 2.7. Clonogenicity Assay

The anti-proliferative effect of QT or QT-SLNs on MCF-7 and MCF-10A cells was measured by a colony formation assessment [39]. Briefly, 3000 cells seeded into 6-well plates and treated with QT or QT-SLNs for 48 h. Afterward, the cells were washed and further incubated with complete medium (DMEM + 10% FBS + 1% pen/strep) for 10 days. Following this, the cells were stained with 0.1% crystal violet in PBS, and the colonies counted under a light microscope (Leica, Wetzlar, Germany).

### 2.8. Annexin V-FITC/Propidium Iodide Apoptosis Assay

MCF-7 and MCF-10A cells (1 × 10^5^) were cultured in a six-well plate and treated with QT or QT-SLN for 48 h. After treatment, normal, apoptotic and necrotic cells were determined using the Annexin V-FITC/propidium iodide assay kit (V13242, Invitrogen, Carlsbad, CA, USA) according to the manufacturer’s protocol. The cells were trypsinized and centrifuged at 1000 rpm, and the cell pellet was washed with PBS and resuspended in 100 mL of binding buffer. The cells were incubated with two mL Annexin V-FITC for 10 min and stained with two mL propidium iodide (PI). Then, the samples were diluted with 400 mL binding buffer and analyzed with a Flow cytometer (Becton Dickinson, San Jose, CA, USA). The different labeling patterns in the Annexin V/PI analysis identified the different cell populations where the FITC negative and PI negative cells were designated as to viable cells; FITC positive and PI negative as to early apoptotic cells; FITC positive and PI positive as to late apoptotic cells and FITC negative and PI positive as to necrotic cells. The data analysis was performed using WinMDI 2.9 software.

### 2.9. Real-Time Polymerase Chain Reaction

RNeasy Mini kit (Qiagen, Hilden, Germany) was used to isolate RNA from cultured cells according to the manufacturer’s instructions. cDNA was produced from the extracted RNAs using the cDNA synthesis kit based on the manufacturer’s protocol (Fermentas, Burlington, ON, Canada). The sequences for all primers were as follows: GAPDH forward primer, 5′-ACCCAGAAGACTGTGGATGG-3′; GAPDH reverse primer: 5′-TTCTAGACGGCAGGTCAGGT-3′, *Bax* forward primer, 5′-GCTGGACATTGGACTTCCTC-3′; *Bax* reverse primer, 5′-ACCACTGTGACCTGCTCCA-3′; *Bcl-2* forward primer, 5′-GCTGGACATTGGACTTCCTC-3′; *Bcl-2* reverse primer, 5′-GCTGGACATTGGACTTCCTC-3′. PCR amplification was performed in 40 cycles using the following program: 95 °C for 10 min, 95 °C for 15 s, 60 °C for 30 s and 60 °C for 34 s. Expression values corrected for the housekeeping gene *GAPDH*. Data were analyzed using the 2^−ΔΔCt^ method. 

### 2.10. Western Analysis

Treated cells were washed by PBS (pH 7.4) and harvested within radioimmunoprecipitation assay (RIPA) lysis buffer containing protease inhibitors. Protein concentration was determined using a BCA assay kit (Pierce Biotechnology Inc., Rockford, IL, USA). Lysate protein (30 °μg) was separated on 10% SDS-PAGE (Novex, San Diego, CA, USA) and transferred onto PVDF membrane (Millipore, Bedford, MA, USA). Primary (Anti-Bcl-2, anti-Bax, and anti-β-actin) and secondary antibodies obtained from Santa Cruz Biotechnology (Santa Cruz, CA, USA). The specific proteins were visualized using an ECL detection kit (Millipore, Burlington, MA, USA). Band density was quantitated using Image J software (National Institutes of Health, Bethesda, MD, USA).

### 2.11. Determination MDA Contents and Antioxidant Enzyme Activities

The MCF-7 and MCF-10A cells treated with Blank-SLN, QT, and QT-SLN for 48 h. Then, the collected samples were lysed, and the protein content of MCF-7 cells and MCF-10A was determined by a BCA protein assay kit (Pierce Biotechnology Inc., Waltham, MA, USA). After centrifugation, the cell lysates, MDA content and CAT and SOD activities were assessed based on the kit’s manufacture (ZellBio, GmbH, Ulm, Germany).

### 2.12. Determination of Intracellular ROS Levels

Levels of ROS were measured using a dichlorodihydrofluorescein diacetate (DCFH-DA) detection kit (Sigma, St. Louis, MO, USA) according to the manufacturer’s instructions. The MCF-7 and MCF-10A cells were seeded in 96-well plates at a density of 5 × 10^3^. After treatment, the medium was removed and the cells were incubated with 10 µM of DCFH-DA plus 100 µL of Hank’s buffered salt solution (HBSS) for 30 min at 37 °C. The levels of ROS were measured using a spectrofluorometer (LS50B, Waltham, MA, USA; Ex: 490 nm, Em: 570 nm).

### 2.13. Statistical Analysis

Data analysis was performed in SPSS (version 21.0, Chicago, IL, USA) using one-way analysis of variance (ANOVA), followed by posthoc pairwise comparison using the Bonferroni procedure. Furthermore, *p*-value of less than 0.05 was considered statistically significant.

## 3. Results

### 3.1. Characterization of QT-SLNs

All formulations showed a negative zeta potential which was in the range of −1.1 to −23.6 mV and the EE% was in the range of 67.6% to 98.9%. The best drug loaded formulation was QT-SLN4, having average particle size 85.5 ± 8.5 nm, zeta potential −22.5 ± 0.6, and PDI 0.152 ± 0.04 (Table 3 and Appendix A).

The mean particle size of QT-SLNs slightly increased in comparison to the blank SLNs (Table 3). This might be a result of the encapsulation of free QT into SLNs. In TEM micrographs, the lipid layer of the SLN had a pale ring around the internal aqueous media, and the QT-SLNs were discrete and had a regular spherical shape (Figure 1). The average particle size given by TEM (88.6 ± 7.9) was in line with that found using DLS, and most of the particles had sizes of less than 100 nm. The zeta potential value of QT-SLNs was high enough to make the nanoparticles repel each other, thus avoiding particle aggregation and keeping the long-term stability of nanoparticles.

The release profile in vitro showed an initial burst release within 0.5 to 6 h and then exhibited a slow QT release. This release of QT-SLNs lasted for 48 h. These finding indicated that the QT-SLNs could provide a slow release of QT during treatment. The released profile in various formulations was similar. The in vitro cumulative percentage of QT release of the QT-SLN4 formulation (Table 1) is illustrated in Figure 2. The QT-SLNs were completely dispersed in aqueous media with no aggregates, whereas free QT exhibited poor aqueous solubility (Appendix A).

### 3.2. Cell Viability and Proliferation

To compare the inhibitory effects of QT on the growth of MCF-7 cells, IC_50_ values were measured. The IC_50_ results for 12, 24, 48 and 72 h of exposure are reported in Table 2. The IC_50_ of QT-SLNs was significantly lower than that of free QT at different times. There was no significant difference between the IC_50_ of free QT and QT-SLNs in 48 h and 72 h. Therefore, the IC_50_ concentration of QT-SLNs in 48 h (25 µmol/mL) was considered to compare the toxic effects of QT and QT-SLNs on MCF-7 cells. According to the results (Figure 3), QT did not significantly reduce viability percentage and colony numbers compared to control cells. The cell viability percentage and colony numbers significantly decreased in QT-SLNs exposed MCF-7 cells, compared to control or QT-treated cells (*p* < 0.01). The cell viability and proliferation of MCF-7 cells were not affected by the Blank-SLN. On the other hand, QT-SLN had no significant effect on viability percentage and colony formation of MCF-10A cells (Table 4).

### 3.3. Morphology Evaluation

In the control group, a few numbers of MCF-7 cells exhibited round morphology. The morphology of QT treated cells was similar to the control group. In the QT-SLNs-treated cells, a large number of cells showed apoptotic morphology, including round shape, cell membrane blebbing and nucleus condensation (Figure 3). Blank-SLN had no significant impact on the morphology of MCF-7 cells. The morphology of the MCF-10A cells was not affected by QT or QT-SLNs (Appendix A).

### 3.4. Annexin V-FITC/Propidium Iodide Apoptosis Assay

In QT exposed MCF-7 cells, apoptotic and necrotic indexes were slightly greater than in control cells. QT-SLNs significantly increased the percentage of late apoptosis, early apoptosis, and necrosis in MCF-7 cells compared to the free QT group (*p* < 0.05). The apoptotic and necrotic indexes of MCF-7 cells were not affected by the Blank-SLN (Figure 4). Apoptotic and necrotic indexes of MCF-10A cells were not significantly altered in response to the QT or QT-SLNs (Table 4).

### 3.5. Quantitative Real-Time RT-PCR

In the QT exposed MCF-7 cells, expression of the *Bax* slightly increased, while the expression of the *Bcl-2* slightly decreased compared to the control. While there was a significant increase in the expression of *Bax* gene, a significant reduction was observed in the expression of Bcl-2 gene in QT-SLN-treated cells compared to the control and QT-treated cells (*p* < 0.01). In the Blank-SLNs group, the expression of *Bax* and *Bcl-2* genes was similar to the control (Figure 5). Gene expression in the MCF-10A cells was not significantly affected by the QT or QT-SLNs (Appendix A).

### 3.6. Western Analysis

No differences in the expression of Bax and Bcl-2 proteins were found between untreated cells and Blank-SLN-treated cells. In the free QT exposed MCF-7 cells, expression of the Bax protein slightly increased, while the expression of the Bcl-2 protein slightly decreased compared to the control. There was a significant increase in the expression of Bax protein, and a significant reduction in the expression of Bcl-2 protein in QT-SLN-treated cells, compared with the control and QT-treated cells (Figure 6). The expression of Bax and Bcl-2 proteins were not significantly affected by the QT or QT-SLNs in the MCF-10A cells (results not shown).

### 3.7. ROS Levels, MDA Content and Antioxidant Enzyme Activity

In the Blank-SLN group, ROS level, MDA content, and SOD and CAT activity were similar to the control. In the free QT-treated cells, ROS level, MDA content and antioxidant enzyme activity slightly changed in comparison with the control group. In the QT-SLN-treated cells, ROS level and MDA content significantly increased in the MCF-7 cells (*p* < 0.01). While SOD and CAT enzyme activity significantly reduced in comparison with control and free QT groups (*p* < 0.05) (Figure 7). The ROS level, MDA content and antioxidant enzyme activity were not significantly affected by QT or QT-SLN in the MCF-10A cells (Table 5).

## 4. Discussion

In this study, QT was successfully incorporated into SLNs through a microemulsification technique. The QT-SLN exhibited a uniform size distribution with excellent stability. The release profile in vitro showed that the QT-SLNs had a slow release of QT during the treatment. The slow release of QT from SLN is suitable for delayed drug release in chemotherapy of breast cancers.

In the present study, we have demonstrated that QT-SLN is effective in reducing cell numbers of MCF-7 cells through growth suppression and inducing cell death. Previous studies showed that QT reduced MCF-7 cell viability [40,41,42]. In this study, QT with an IC_50_ of 41.5 µmol for 48 h could prevent the growth of MCF-7 cancer cells. Inconsistent with our results, Lin et al. (2008) have also reported that QT, at a dose of 40 µmol, much reduced the number of MCF-7 cells [40]. Dhumale et al. (2015) and Li et al. (2018) showed that QT with an IC_50_ of 50 µmol effectively prevents the growth of breast cancer cells [14,41]. The IC_50_ of QT-SLNs (25 µmol) was markedly lower than the free QT, indicating the greater toxicity of QT-loaded nanoparticles on MCF-7 cells. The viability percentage and colony formation were lower than that of free QT, indicated that the QT-SLN was a good delivery system for the breast cancer cells and more effective than QT alone toward MCF7 cells. Various research studies confirm that SLN is a suitable carrier for anticancer drugs and phytochemical components. The encapsulation of Berberine, Oridonin, and Resveratrol in SLNs have shown an enhanced antitumor effect in MCF-7 cell lines [21,33,34]. Zhuang et al. (2012) reported that anticancer drugs encapsulated in SLN, including paclitaxel, mitoxantrone, and methotrexate, may be more effective than free anticancer drugs for breast cancer treatment [42]. In contrast, Abbasalipourkabir et al. (2016) showed that tamoxifen-loaded SLNs had a similar effect on rat breast tumor to free tamoxifen [43].

The enhanced preventive effect of QT-SLNs on the growth of MCF-7 cells may be related to the lipophilic property of the carrier, which promotes the intracellular uptake. In the study of Vijayakumar et al., QT-loaded SLN worked better than free QT in distilled water, and SLN greatly increased cellular uptake of QT [36]. Sun et al. (2014) demonstrated that QT-nanostructured lipid carriers markedly enhanced the QT solubility and stability and increased the QT content in MCF-7 cells. In their study, the enhanced cytotoxicity was parallel to increased QT uptake by MCF-7 cells [44].

Cell survival and apoptosis are often applied to check the efficacy of anti-cancer agents. Anticancer drugs generally kill dividing cells by activation of the apoptosis process [45,46,47]. We examined the ability of QT-SLN treatment to induce apoptosis in MCF-7 cells by flow cytometry and gene expression. The significantly increased apoptosis suggested that the reduction of MCF-7 cell viability after QT-SLN treatment was due to the stimulation of apoptosis. Consistent with our results, Jain et al. (2014) showed that QT-loaded self-nanoemulsifying induces DNA damage and apoptosis in MCF-7 cells [48].

Morphology assessment confirmed the flow cytometry results and was consistent with the viability percentage and colony formation of the control and experimental groups.

In this study, QT-SLNs markedly enhanced *Bax* expression, since there was a significant reduction in *Bcl-2* expression. *Bcl-2* and *Bax* proteins, the main members of the *Bcl-2* family, regulate the intrinsic pathway of apoptosis [49]. Therefore, QT-SLNs can stimulate the intrinsic apoptotic pathway to induce cell death in MCF-7 cancer cells. Imbalance of the *Bax*/*Bcl-2* ratio can change tumor cells sensitivity to cell death induced by chemotherapeutic drugs or radiation [50].

In the study of Lee et al. (2008), *Bax* involved in the QT-induced apoptosis in prostate cancer cells [51]. QT induced apoptosis and suppressed the growth of MCF-7 cells by regulating the expression of *Bax* and *Bcl-2* [52].

As shown in the results, QT-SLN effectively increased ROS levels and MDA content, and reduced SOD and CAT activity in the MCF-7 cells. Hence, QT-SLN may show pro-oxidant activity in breast cancer cells. Pro-oxidant action of tea polyphenols links to their anticancer actions [8]. Morinda citrifolia (Noni) alters oxidative stress marker, MDA content and antioxidant activity (SOD and CAT) in the cervical cancer cell lines [9].

It has been reported that raised levels of reactive oxygen species (ROS) can regulate the expression of Bcl-2 family proteins and induce apoptosis [53]. In the study of Ren et al. (2018), paclitaxel (a diterpenoid compound) could elevate ROS and MDA levels, whereas SOD activity decreased in mammary gland tumors. In their study, paclitaxel-induced apoptosis by downregulation of Bcl-2 and upregulation of Bax proteins [54]. Farnesiferol C (isolated from *Ferula asafoetida*) significantly induced apoptosis by increasing the cellular ROS levels in the MCF-7 cells [55]. By contrast, in the study by Yao et al., curcumin-induced apoptosis was accompanied by reducing ROS and MDA levels and increased SOD activity in lung cancer cells (A549 cells) [56].

Flow cytometry results show that the percentage of necrosis was markedly enhanced in QT-SLN exposed MCF-7 cells, which confirms that QT-SLN can stimulate multiple cell death pathways. In a previous study, QT was able to induce apoptosis and necrosis in SCC-9 oral cancer cells [57].

Apoptosis and necrosis were mediated by distinct but overlapping pathways involving mitochondria/endoplasmic reticulum [58]. Melatonin was involved in the necrosis and apoptosis of the pancreatic cancer cell line SW-1990 by modulating of the *Bax*/*Bcl-2* ratio [59]. Whelan et al. (2012) found that deletion of *Bax* significantly reduces necrotic injury during myocardial infarction [58]. Therefore, QT-SLN is able to stimulate both apoptotic and necrotic pathways by modulating the *Bax*/*Bcl-2* ratio.

In this study, the non-tumorigenic MCF-10A cells were not affected by QT-SLNs. Thus, QT-SLNs have great potential in adjuvant therapy for clinical application in breast cancer. In agreement with our findings, apoptosis and decreasing viability percentage induced in several cancer cell lines [60,61], but not in nontumoral MCF-10A cells [62] after treatment with QT. Cancer cells are more susceptible to being killed by anticancer flavonoids compared to normal cells. It has been reported that the same dose of flavonoids induces apoptosis in cancer cells, but not in their normal counterparts [63,64].

## 5. Conclusions

This study demonstrated that SLN effectively increased cytotoxic effects of QT by inducing oxidative stress and stimulating the intrinsic pathway of apoptosis in MCF-7 cells. Therefore, QT-loaded SLN may be of use in the treatment breast cancer in the future. More studies are required to increase our knowledge of cell death signaling pathways in QT-SLN-treated cancer cells.

## Figures and Tables

**Figure 1 medicina-55-00114-f001:**
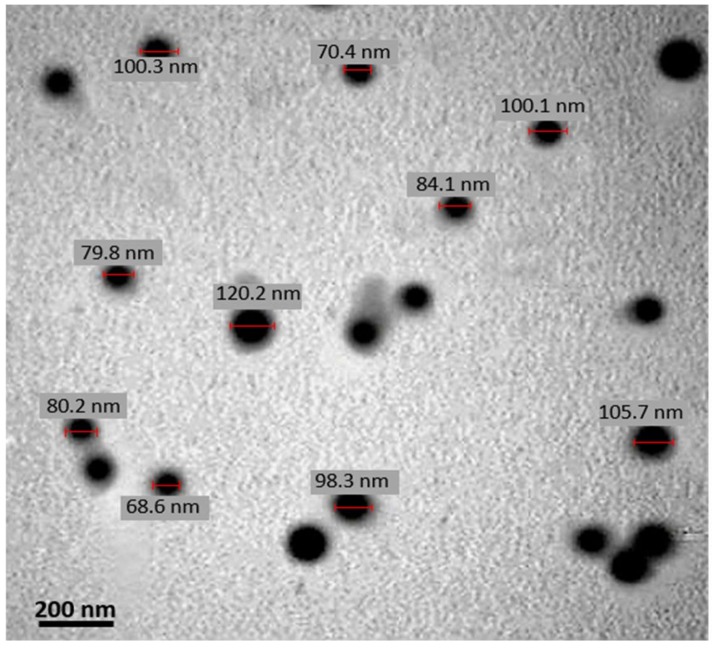
A TEM micrograph of QT-SLNs. The lipid layer of the SLN can be observed as pale rings around the internal aqueous media. Scale bar: 200 nm.

**Figure 2 medicina-55-00114-f002:**
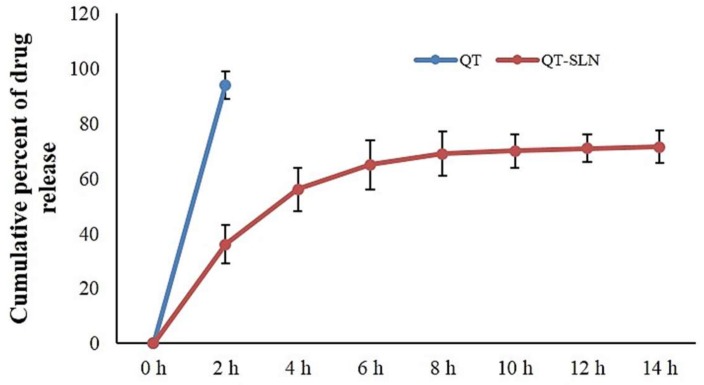
In vitro cumulative percentage of drug release vs. time. Data expressed as mean ± SD (*n* = 6).

**Figure 3 medicina-55-00114-f003:**
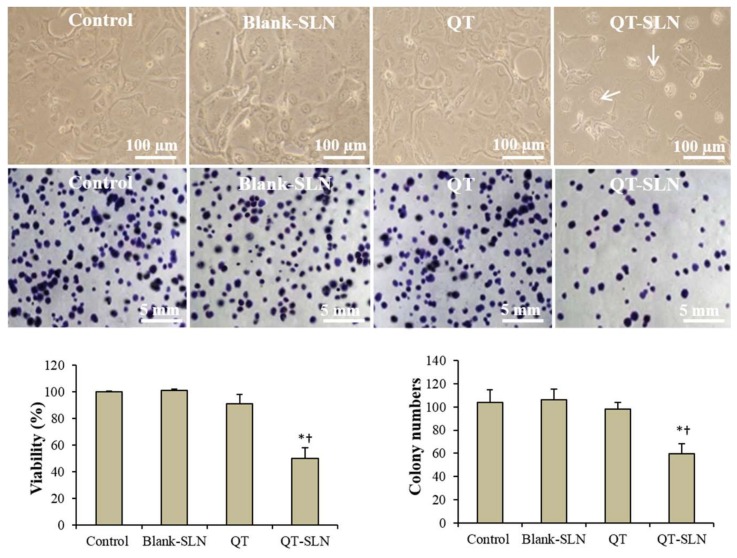
Morphology (upper pictures) and colony formation (lower pictures) of MCF-7 cells in the control and experimental groups. Arrows indicate apoptotic morphology. Viability and colony numbers of control and experimental groups are also observed. Values are expressed as mean ± SD. * *p* < 0.01, ^†^
*p* < 0.01; * and ^†^ symbols respectively indicate comparison to control and QT-treated cells.

**Figure 4 medicina-55-00114-f004:**
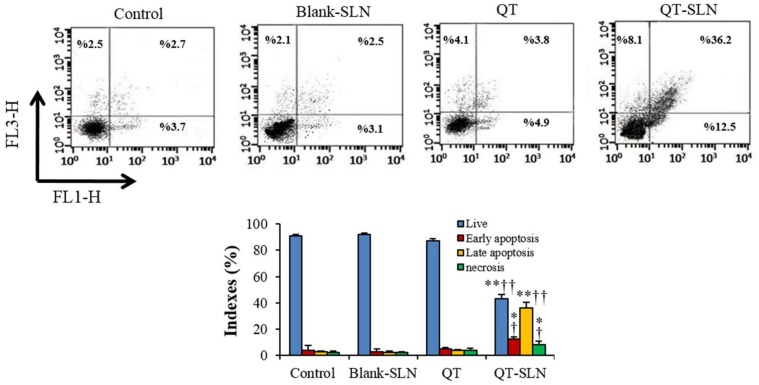
Flow cytometry of Annexin/PI staining in control and experimental groups. Lower left quadrant: live cells; lower right quadrant: early apoptosis; upper right quadrant: late apoptosis; upper left quadrant: necrotic cells. All assays were performed in triplicate, and the mean ± standard deviations are shown. * *p* < 0.05, ** *p* < 0.001, ^†^
*p* < 0.01, ^††^
*p* < 0.001; * and ^†^ symbols respectively indicate the comparison to control and QT groups.

**Figure 5 medicina-55-00114-f005:**
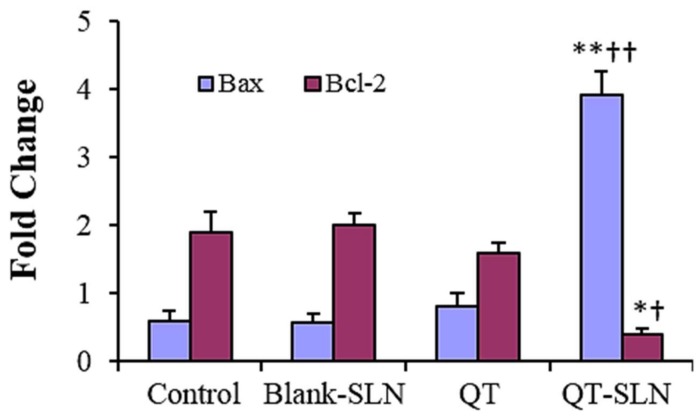
Gene expression for MCF-7 cells in different groups. Expression is normalized to average of housekeeping gene (*GAPDH*). Values are expressed as mean ± SD. * *p* < 0.01, ** *p* < 0.001, ^†^
*p* < 0.01, ^††^
*p* < 0.001; * and ^†^ symbols respectively indicate the comparison to control and QT-treated cells.

**Figure 6 medicina-55-00114-f006:**
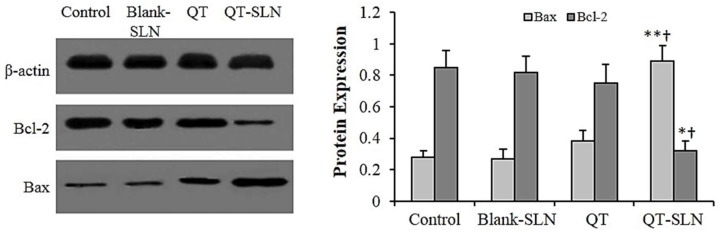
The level of protein expression in the MCF-7 cells. The β-actin was used as the loading control. The protein levels of Bax and Bcl-2 in the control and treated cells were quantified by ImageJ software and normalized to β-actin band intensity. Values are expressed as mean ± SD. * *p* < 0.05, ** *p* < 0.01, ^†^
*p* < 0.05; * and ^†^ symbols respectively indicate the comparison to control and QT-treated cells.

**Figure 7 medicina-55-00114-f007:**
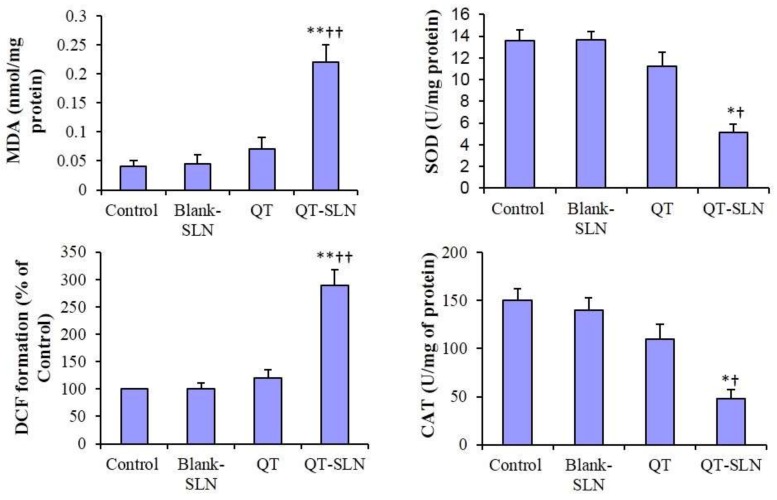
MDA content, ROS levels (DCF formation) and antioxidant enzyme activity in the MCF-7 cells. Values are expressed as mean ± SD. * *p* < 0.05, ** *p* < 0.00, ^†^
*p* < 0.05, ^††^
*p* < 0.01; * and ^†^ symbols respectively indicate the comparison to control and QT-treated cells.

**Table 1 medicina-55-00114-t001:** Formulations of QT-SLN.

Formulation	Drug-Lipid Ratio	QT (mg)	Compritol (mg)	Tween 80 (mL)	EE (%)	LD (%)
QT-SLN1	1:1	50	50	6	67.6 ± 4.3	19.7 ± 2.6
Blank-SLN1	-	-	50	6	-	-
QT-SLN2	1:3	50	150	6	78.4 ± 4.8	21.5 ± 2.9
Blank-SLN2	-	-	150	6	-	-
QT-SLN3	1:5	50	250	6	89.3 ± 5.8	27.3 ± 3.9
Blank-SLN3	-	-	250	6	-	-
QT-SLN4	1:10	50	500	6	97.6 ± 2.3	28.3 ± 3.8
Blank-SLN4	-	-	500	6	-	-
QT-SLN5	1:15	50	750	6	98.9 ± 1.1	28.6 ± 4.1
Blank-SLN5	-	-	750	6	-	-

Results are given as mean ± SD (*n* = 3). SD: standard deviation, LD: loading drug, EE: encapsulation efficiency.

**Table 2 medicina-55-00114-t002:** The IC_50_ (µmol/ mL) of QT, QT-SLN, and SLN on MCF-7 and MCF-10A cells after different exposure times.

Treatments	Cells	12 h	24 h	48 h	72 h
QT	MCF-7 MCF-10A	86.7 ± 7.5 218.3 ± 16.9	73.8 ± 7.1 201.2 ± 16.3	41.5 ± 6.1 **^†^ 178.4 ± 14.6 *	40.2 ± 5.8 **^†^ 174.4 ± 15.1 *
QT-SLN	MCF-7 MCF-10A	48.8 ± 4.3 198.5 ± 13.2	36.7 ± 3.5 * 182 ± 11.1	25.01 ± 2.4 **^†^ 176.8 ± 12.5	24.7 ± 2.7 **^†^ 171.3 ± 10.8
Blank-SLN	MCF-7 MCF-10A	293.2 ± 18.4 288.3 ± 17.8	289.7 ± 18.1 283.4 ± 19.1	284.5 ± 16.4 282.7 ± 16.9	281.4 ± 16.8 278.6 ± 18.2

Values are expressed as mean ± SD (*n* = 5). * *p* < 0.05, ** *p* < 0.01, ^†^
*p* < 0.05; * and ^†^ symbols respectively indicate comparison to 12 and 24 h.

**Table 3 medicina-55-00114-t003:** Characteristics of the formulation of QT/Blank-SLNs.

Formulation	Drug-Lipid Ratio	Particle Size	PDI	Zeta Potential (mV)
QT-SLN1	1:1	45.5 ± 3.5	0.112 ± 0.01	−1.8 ± 0.26
Blank-SLN1	-	45.1 ± 3.2	0.114 ± 0.02	−1.1 ± 0.35
QT-SLN2	1:3	48.4 ± 3.9	0.118 ± 0.07	−5.5 ± 1.12
Blank-SLN2	-	47.9 ± 4.1	0.123 ± 0.06	−8.9 ± 1.33
QT-SLN3	1:5	58.3 ± 4.8	0.135 ± 0.07	−12.6 ± 2.32
Blank-SLN3	-	56.1 ± 4.6	0.127 ± 0.06	−13.9 ± 1.87
QT-SLN4	1:10	85.5 ± 8.5	0.152 ± 0.04	−22.5 ± 0.6
Blank-SLN4	-	84.7 ± 8.1	0.161 ± 0.05	−23.6 ± 0.5
QT-SLN5	1:15	99.6 ± 9.1	0.342 ± 0.04	−18.9 ± 3.13
Blank-SLN5	-	98.8 ± 8.7	0.316 ± 0.11	−20.3 ± 2.58

Results are given as mean ± SD (*n* = 3). SD: standard deviation, PDI: polydispersity index.

**Table 4 medicina-55-00114-t004:** The results obtained from MTT assay, colony formation and Annexin V/PI method in MCF-10A cells.

Groups	Control	Blank SLN	QT	QT-SLNs
Viability (%)	100.0 ± 0.02	100.1 ± 0.09	101.2 ± 1.8	104.3 ± 2.1
Colony numbers (%)	1.42 ± 0.14	1.29 ± 0.22	1.38 ± 0.18	1.43 ± 0.12
Early apoptosis (%)	3.12 ± 0.56	2.96 ± 0.32	3.14 ± 0.62	2.87 ± 0.35
Late apoptosis (%)	2.57 ± 0.25	2.61 ± 0.25	2.26 ± 0.23	2.09 ± 0.15
Necrosis (%)	1.27 ± 0.08	1.01 ± 0.13	1.23 ± 0.05	1.26 ± 0.04

Values are expressed as mean ± SD (*n* = 6).

**Table 5 medicina-55-00114-t005:** The oxidative stress results in the MCF-10A cells.

Groups	Control	Blank SLN	QT	QT-SLNs
MDA (nmol/mg protein)	0.042 ± 0.00	0.043 ± 0.00	0.041 ± 0.00	0.036 ± 0.00
CAT (U/mg protein)	130.3 ± 0.00	129.4 ± 6.51	131.7 ± 8.34	135.5 ± 9.13
SOD (U/mg protein)	13.71 ± 3.21	13.63 ± 3.08	13.86 ± 3.68	13.95 ± 4.05
ROS (% of control)	100 ± 0.00	99.72 ± 0.21	99.84 ± 0.31	99.47 ± 0.29

Values are expressed as mean ± SD (*n* = 6).

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
