# Peer review of "Effects of Quercetin-Loaded Nanoparticles on MCF-7 Human Breast Cancer Cells"

_medicina, 2019, doi:10.3390/medicina55040114_

Round 1

Reviewer 1 Report

This manuscript describes the encapsulation of Quercetin in QT-SLNs. Although it is a well-performed study there are some points that could be revised.

The minor changes regard:

Line 19 (abstract): please remove the term “healthy cells” and use “non-tumorigenic cell line”. This is the most correct definition for MCF-10A cells.

Lines 38-39 (introduction): in this section, not only herbal medicine but also drug-targeting therapies are useful strategies to treat cancer, especially breast one, not only for triple-negative. Please add references, for example:

Wei Lv et al., 2016 J Med Chem 59,10, 4511-4525.

F. Aiello et al., 2017, ChemMedChem 12,16, 1279-1285.

Lines 302-303: please add author contributions

Lines 304-305: if present, please add your funding sources.

On the other hand, I have some requests:

-          The MCF-7 cell line regards a breast cancer that is strongly treated by common chemotherapeutic agents. Why did you use only this cell line and not only for example, the most aggressive triple negative breast cancer MDA-MB231 cell line or also the less metastatic MDA-MB468 or the GPR30 responsive SkBr3, which are the most similar to human breast cancer tissues. The data regarding this cell line should enrich the already high quality of the manuscript. Are you able to perform these experiments in short time?

-          Regarding the biological data reported, why the nanoparticle is ale to reduce, when “free of QT” the viability of MCF-10A?

-          Usually, as you reported, the anti-proliferative properties of QT are limited by its autooxidation or more over by its metabolism. Why did not you perform an antioxidant assay to the nanoparticle, or also why did not you use the ROS assay on cell line?

After this revision, I will be happy to approve the publication of this interesting manuscript.

Author Response

The minor changes regard:

Line 19 (abstract): please remove the term “healthy cells” and use “non-tumorigenic cell line”. This is the most correct definition for MCF-10A cells.

Response: It is corrected.

Lines 38-39 (introduction): in this section, not only herbal medicine but also drug-targeting therapies are useful strategies to treat cancer, especially breast one, not only for triple-negative. Please add references, for example:

Wei Lv et al., 2016 J Med Chem 59,10, 4511-4525.

F. Aiello et al., 2017, ChemMedChem 12,16, 1279-1285.

Response: It is revised. New references are added to the text.

Lines 302-303: please add author contributions

Response: It is revised.

Lines 304-305: if present, please add your funding sources.

Response: It is revised.

On the other hand, I have some requests:

-          The MCF-7 cell line regards a breast cancer that is strongly treated by common chemotherapeutic agents. Why did you use only this cell line and not only for example, the most aggressive triple negative breast cancer MDA-MB231 cell line or also the less metastatic MDA-MB468 or the GPR30 responsive SkBr3, which are the most similar to human breast cancer tissues. The data regarding this cell line should enrich the already high quality of the manuscript. Are you able to perform these experiments in short time?

Response: There are a large number of published articles in which the effects of different compounds studied only on MCF-7 cells. Please, note the references 41, 55 and 62 of the present article. We were unable to perform these experiments in a short time. We will consider your worth comment in our future studies.

-          Regarding the biological data reported, why the nanoparticle is ale to reduce, when “free of QT” the viability of MCF-10A?

Response: Unfortunately, I did not understand you. There was no significant difference in the viability of the MCF-10A cells in the different groups. The viability of the QT-SLN-treated MCF-10A cells were slightly more than the QT-treated cells. This finding may relate to much antioxidant activity of the QT-SLN.

-          Usually, as you reported, the anti-proliferative properties of QT are limited by its autooxidation or more over by its metabolism. Why did not you perform an antioxidant assay to the nanoparticle, or also why did not you use the ROS assay on cell line?

Response: We measured the ROS levels, MDA content, and SOD and CAT activity. The data added to the text and discussed. Please, note to the abstract, introduction, methods, results and discussion.

Reviewer 2 Report

This study investigated the cytotoxic effects of QT-SLNs on MCF-7, one of representative human breast cancer cell line. The authors demonstrated that Solid Lipid Nano capsulation effectively increased cytotoxic effects of quercetin and the mechanism is associated with the intrinsic pathway of apoptosis including upregulation of Bax in MCF-7 cells. Therefore, the authors suggest that SLNs can be a suitable carrier for anticancer drugs and phytochemical components. Most of the data is clear but this reviewer has a couple of suggestions.

Major issues

Data from figure 2 is not clear. If the authors want to say QT-SLNs could provide a slow release of QT during treatment.” and “the QT-SLNs were completely dispersed in aqueous media with no aggregates”, please provide additional data (in vitro release assay with free QT) to compare with data from SLN-QT.

Generally, protein expression (by western blot) is more conclusive than RNA expression. Is there any reason to examine mRNA expression rather than protein expression of Bcl or Bax?

Minor issues

More References are required for line 41 QT has been found to be very effective against many types of cancer cells’

Reference format is different in line 45 “Several drug delivery systems, such as polymeric nanoparticles, solid lipid nanoparticles (SLNs), liposomes, and micelles, have been examined in order to increase the targeting ability and the bioavailability of anticancer agents (10-13)” and several same mistakes exist in discussion section, too.

Written English is poor; English editing seems to be required

Author Contributions and Funding should be properly demonstrated.

Author Response

Dear editor

The reviewer's comments considered and the corrections highlighted (yellow color) in the main text.

Reviewer 2

Major issues

Data from figure 2 is not clear. If the authors want to say “QT-SLNs could provide a slow release of QT during treatment.” and “the QT-SLNs were completely dispersed in aqueous media with no aggregates”, please provide additional data (in vitro release assay with free QT) to compare with data from SLN-QT.

Response: it is revised. In vitro release assay with free QT depicted in figure 2, and a figure for showing completely dispersion of QT-SLNs in aqueous media added to the supplemental file.

Generally, protein expression (by western blot) is more conclusive than RNA expression. Is there any reason to examine mRNA expression rather than protein expression of Bcl or Bax?

Response: We examined protein expression of Bcl and Bax by western blot. And the data added to the text. 

Minor issues

More References are required for line 41 “QT has been found to be very effective against many types of cancer cells’

Response: it is revised.

Reference format is different in line 45 “Several drug delivery systems, such as polymeric nanoparticles, solid lipid nanoparticles (SLNs), liposomes, and micelles, have been examined in order to increase the targeting ability and the bioavailability of anticancer agents (10-13)” and several same mistakes exist in discussion section, too.

Response: it is revised.

The manuscript was checked by a native English speaking colleague.

“Author Contributions” and “Funding” should be properly demonstrated.

Response: it is revised.

Reviewer 3 Report

The manuscript is well constructed and described in vitro cytotoxicity of quercetin against MCF-7 human breast cancer cells. The introduction, materials and methods as well as the results are presented clearly with the excellent figures put them clearly into content. My major concern is about enhancing the cytotoxicity of quercetin through solid liquid nanoparticles will be shown or investigated in bioavailability assay model, which will make more strength of the in vitro experimental work. Moreover, there are few minor issues requires to be corrected.

Page 3; Line 83: What was the concentration of methanol in percentage?

Page 3; Line 88: Need reference for the method.

Page 6; Line 187: The IC50 results were showed in table 2, not in table 3.

Page 6; Line 189-190: If the IC50 of free QT and QT-SLNs in 48 h was more than 72 h, the it should be considered less active. Then why it has been considered or given priority over the activity after 72 h which has less IC50 value.

Author Response

Dear editor

The reviewer's comments considered and the corrections highlighted (yellow color) in the main text.

Reviewer 3

The manuscript is well constructed and described in vitro cytotoxicity of quercetin against MCF-7 human breast cancer cells. The introduction, materials and methods as well as the results are presented clearly with the excellent figures put them clearly into content. My major concern is about enhancing the cytotoxicity of quercetin through solid liquid nanoparticles will be shown or investigated in bioavailability assay model, which will make more strength of the in vitro experimental work. Moreover, there are few minor issues requires to be corrected.

Response: We will consider your worth comment in our future studies.

Page 3; Line 83: What was the concentration of methanol in percentage?

Response: Absolute methanol was used.

Page 3; Line 88: Need reference for the method.

Response: it is revised.

Page 6; Line 187: The IC50 results were showed in table 2, not in table 3.

Response: it is revised.

Page 6; Line 189-190: If the IC50 of free QT and QT-SLNs in 48 h was more than 72 h, the it should be considered less active. Then why it has been considered or given priority over the activity after 72 h which has less IC50 value.

Response: it is revised. There was no significant difference between the IC50 of free QT and QT-SLNs in 48 h and 72 h. So, IC50 concentration of QT-SLNs in 48 h (25 ?mol/ mL) was considered to compare the toxic effects of QT and QT-SLNs on MCF-7 cells.

Round 2

Reviewer 2 Report

It is appropriately revised accoriding to reviwer's comments.